# GeneFace: Generalized and High-Fidelity Audio-Driven 3D Talking Face Synthesis

**Zhenhui Ye**[1]\*, **Ziyue Jiang**[1]\*, **Yi Ren**[2], **Jinglin Liu**[1], **JinZheng He**[1], **Zhou Zhao**[1]†

[1]School of Computer Science and Technology, Zhejiang University
{zhenhuiye,jiangziyue,jinglinliu,jinzhenghe,zhaozhou}@zju.edu.cn

[2]ByteDance
ren.yi@bytedance.com

## Abstract

Generating photo-realistic video portraits with arbitrary speech audio is a crucial problem in film-making and virtual reality. Recently, several works explore the usage of neural radiance field (NeRF) in this task to improve 3D realness and image fidelity. However, the generalizability of previous NeRF-based methods to out-of-domain audio is limited by the small scale of training data. In this work, we propose GeneFace, a generalized and high-fidelity NeRF-based talking face generation method, which can generate natural results corresponding to various out-of-domain audio. Specifically, we learn a variational motion generator on a large lip-reading corpus, and introduce a domain adaptive post-net to calibrate the result. Moreover, we learn a NeRF-based renderer conditioned on the predicted facial motion. A head-aware torso-NeRF is proposed to eliminate the head-torso separation problem. Extensive experiments show that our method achieves more generalized and high-fidelity talking face generation compared to previous methods[1].

## 1 Introduction

Audio-driven face video synthesis is an important and challenging problem with several applications such as digital humans, virtual reality (VR), and online meetings. Over the past few years, the community has exploited Generative Adversarial Networks (GAN) as the neural renderer and promoted the frontier from only predicting the lip movement (Prajwal et al., 2020; Chen et al., 2019) to generating the whole face (Zhou et al., 2021; Lu et al., 2021). However, GAN-based renderers suffer from several limitations such as unstable training, mode collapse, difficulty in modelling delicate details (Suwajanakorn et al., 2017; Thies et al., 2020), and fixed static head pose (Pham et al., 2017; Taylor et al., 2017; Cudeiro et al., 2019). Recently, Neural Radiance Field (NeRF) (Mildenhall et al., 2020) has been explored in talking face generation. Compared with GAN-based rendering techniques, NeRF renderers could preserve more details and provide better 3D naturalness since it models a continuous 3D scene in the hidden space.

Recent NeRF-based works (Guo et al., 2021; Liu et al., 2022; Yao et al., 2022) manage to learn an end-to-end audio-driven talking face system with only a few-minutes-long video. However, the current end-to-end framework is faced with two challenges. 1) The first challenge is the **weak generalizability** due to the small scale of training data, which only consists of about thousands-many audio-image pairs. This deficiency of training data makes the trained model not robust to out-of-domain (OOD) audio in many applications (such as cross-lingual (Guo et al., 2021; Liu et al., 2022) or singing voice). 2) The second challenge is the so-called **"mean face" problem**. Note that the audio to its corresponding facial motion is a one-to-many mapping, which means the same audio input may have several correct motion patterns. Learning such a mapping with a regression-based model

---

\*Authors contribute equally to this work.
†Corresponding author
[1]Video samples and source code are available at https://geneface.github.io

leads to over-smoothing and blurry results (Ren et al., 2021); specifically, for some complicated audio with several potential outputs, it tends to generate an image with a half-opened and blurry mouth, which leads to unsatisfying image quality and bad lip-synchronization. To summarize, the current NeRF-based methods are challenged with the weak generalizability problem due to the lack of audio-to-motion training data and the "mean face" results due to the one-to-many mapping.

In this work, we develop a talking face generation system called **GeneFace** to address these two challenges. To handle the weak generalizability problem, we devise an audio-to-motion model to predict the 3D facial landmark given the input audio. We utilize hundreds of hours of audio-motion pairs from a large-scale lip reading dataset (Afouras et al., 2018) to learn a robust mapping. As for the "mean face" problem, instead of using the regression-based model, we adopt a variational auto-encoder (VAE) with a flow-based prior as the architecture of the audio-to-motion model, which helps generate accurate and expressive facial motions. However, due to the domain shift between the generated landmarks (in the multi-speaker domain) and the training set of NeRF (in the target person domain), we found that the NeRF-based renderer fails to generate high-fidelity frames given the predicted landmarks. Therefore, a domain adaptation process is proposed to rig the predicted landmarks into the target person's distribution. To summarize, our system consists of three stages:

1 **Audio-to-motion**. We present a variational motion generator to generate accurate and expressive facial landmark given the input audio.

2 **Motion domain adaptation**. To overcome the domain shift, we propose a semi-supervised adversarial training pipeline to train a domain adaptive post-net, which refines the predicted 3D landmark from the multi-speaker domain into the target person domain.

3 **Motion-to-image.** We design a NeRF-based renderer to render high-fidelity frames conditioned on the predicted 3D landmark.

The main contributions of this paper are summarized as follows:

- We present a three-stage framework that enables the NeRF-based talking face system to enjoy the large-scale lip-reading corpus and achieve high generalizability to various OOD audio. We propose an adversarial domain adaptation pipeline to bridge the domain gap between the large corpus and the target person video.

- We are the first work that analyzes the "mean face" problem induced by the one-to-many audio-to-motion mapping in the talking face generation task. To handle this problem, we design a variational motion generator to generate accurate facial landmarks with rich details and expressiveness.

- Experiments show that our GeneFace outperforms other state-of-the-art GAN-based and NeRF-based baselines from the perspective of objective and subjective metrics.

## 2 RELATED WORK

Our approach is a 3D talking face system that utilizes a generative model to predict the 3DMM-based motion representation given the driving audio and employs a neural radiance field to render the corresponding images of a human head. It is related to recent approaches to audio-driven talking head generation and scene representation networks for the human portrait.

**Audio-driven Talking Head Generation** With the progress of audio synthesis technology (Ye et al., 2022; Huang et al., 2022a;b;c) and generative models(Zhang et al., 2022; 2023), generating talking faces in line with input audio has attracted much attention of the computer vision community. Earlier works focus on synthesizing the lip motions on a static facial image (Jamaludin et al., 2019; Tony Ezzat & Poggio, 2002; Vougioukas et al., 2020). Then the frontier is promoted to synthesize the full head (Yu et al., 2020; Zhou et al., 2019; 2020). However, free pose control is not feasible in these methods due to the lack of 3D modeling. With the development of 3D face reconstruction techniques (Deng et al., 2019), many works explore extracting 3D Morphable Model (3DMM) (Paysan et al., 2009) from the monocular video to represent the facial movement (Tero Karras & Lehtinen, 2017; Yi et al., 2020) in the talking face system, which is named as model-based methods. With 3DMM, a coarse 3D face mesh $M$ can be represented as an affine model of facial expression and identity code:

$$M = \overline{M} + B_{id}\mathbf{i} + B_{exp}\mathbf{e}, \tag{1}$$

Figure 1: The inference process of GeneFace. BN denotes batch normalization.

where $\overline{M}$ is the average face shape; $B_{id}$ and $B_{exp}$ are the PCA bases of identity and expression; $\mathbf{i}$ and $\mathbf{e}$ are known as identity and expression codes.

By modeling the 3D geometry with 3DMM, the model-based works manage to manipulate the head pose and facial movement. However, 3DMM could only define a coarse 3D mesh of the human head, and delicate details (such as hair, wrinkle, teeth, etc.) are ignored. It raises challenges for GAN-based methods to obtain realistic results. Recent advances in neural rendering have created a prospect: instead of refining the geometry of 3DMM or adding more personalized attributes as auxiliary conditions for GAN-based renderers, we could leave these delicate details to be modeled implicitly by the hidden space of the neural radiance field.

**Neural Radiance Field for Rendering Face**  The recent proposed neural radiance field (NeRF) (Mildenhall et al., 2020; Kellnhofer et al., 2021; Pumarola et al., 2021; Sitzmann et al., 2019) has attracted much attention in the human portrait rendering field since it could render high-fidelity images with rich details such as hair and wrinkles. For instance, Sitzmann et al. (2019) presents a compositional NeRF for generating each part of the upper body. Gafni et al. (2021) and Pumarola et al. (2021) propose pose-expression-conditioned dynamic NeRFs for modeling the dynamics of a human face. Chan et al. (2022) proposes a hybrid explicit–implicit tri-plane representation to achieve fast and geometry-aware human face rendering. Hong et al. (2022) proposes a real-time NeRF-based parametric head model.

Several works have also applied NeRF in the audio-driven talking face generation task. Zhang et al. (2021) devises an implicit pose code to modularize audio-visual representations. Guo et al. (2021) first presents an end-to-end audio-driven NeRF to generate face images conditioned on Deepspeech (Hannun et al., 2014) audio features. Recently, Liu et al. (2022) proposed a semantic-aware dynamic ray sampling module to improve the sample efficiency and design a torso deformation module to stabilize the large-scale non-rigid torso motions. Yao et al. (2022) introduces two disentangled representations (eye and mouth) to provide improved conditions for NeRF. To achieve few-shot training, Shen et al. (2022) conditions the face radiance field on 2D reference images to learn the face prior, thus greatly reducing the required data scale (tens of seconds of video) and improve the convergence speed (about 40k iterations). However, all of the previous NeRF-based work focuses on better image quality or reducing the training cost, while the generalizability to out-of-domain audio is relatively an oversight.

Our GeneFace could be regarded as bridging the advantages of the aforementioned two types of works. Compared with previous 3DMM-based methods, our work could enjoy good 3D naturalness and high image quality brought by the NeRF-based renderer. Compared with previous end-to-end NeRF-based methods, we improve the generalizabity to out-of-domain audio via introducing a generative audio-to-motion model trained on a large lip reading corpus.

## 3  GENEFACE

In this section, we introduce our proposed GeneFace. As shown in Fig. 1, GeneFace is composed of three parts: 1) a variational motion generator that transforms HuBERT features (Hsu et al., 2021) into 3D facial landmarks; 2) a post-net to refine the generated motion into the target person domain; 3) a NeRF-based renderer to synthesize high-fidelity frames. We describe the designs and the training process of these three parts in detail in the following subsections.

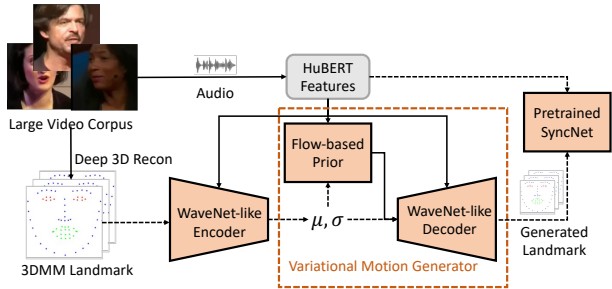

Figure 2: The structure of variational motion generator. Dashed arrows means the process is only performed during training; and only the dashed rectangle part is used during inference.

## 3.1 VARIATIONAL MOTION GENERATOR

To achieve expressive and diverse 3D head motion generation, we introduce a variational auto-encoder (VAE) to perform a generative and expressive audio-to-motion transform, namely the variational motion generator, as shown in Fig. 2.

**Audio and motion representation**  To better extract the semantic information, we utilize Hu-BERT, a state-of-the-art ASR model, to obtain audio features from the input wave and use it as the condition of the variational motion generator. As for the motion representation, to represent detailed facial movement in Euclidean space, we select 68 key points from the reconstructed 3D head mesh and use their position as the action representations. Specifically,

$$LM_{3D} = \{(M - \overline{M})_i | i \in I\}, \tag{2}$$

where $LM_{3D} \in \mathbb{R}^{68 \times 3}$, $M$ and $\overline{M}$ are the 3DMM mesh and mean mesh defined in Equation (1), $I$ is the index of the key landmark in the mesh. In this paper, we name this action representation *3D landmarks* for abbreviation.

**Dilated convolutional encoder and decoder**  Inspired by WaveNet, to better extract features from the audio sequence and construct long-term temporal relationships in the output sample, we design the encoder and decoder as fully convolutional networks where the convolutional layers have incrementally increased dilation factors that allow its receptive field to grow exponentially with depth. In contrast to previous works, which typically divide the input audio sequence into sliding windows to obtain a smooth result, we manage to synthesize the whole sequence of arbitrary length within a single forward. To further improve the temporal stability of the predicted landmark sequence, a Gaussian filter is performed to eliminate tiny fluctuations in the result.

**Flow-based Prior**  We also notice that the gaussian prior of vanilla VAE limits the performance of our 3D landmark sequence generation process from two prospectives: 1) the datapoint of each time index is independent of each other, which induces noise to the sequence generation task where there is a solid temporal correlation among frames. 2) optimizing VAE prior push the posterior distribution towards the mean, limiting diversity and hurting the generative power. To this end, following Ren et al. (2021), we utilize a normalizing flow to provide a complex and time-related distribution as the prior distribution of the VAE. Please refer to Appendix A.1 for more details.

**Training Process**  Due to the introduction of prior flow, the closed-form ELBO is not feasible, hence we use the Monte-Carlo ELBO loss (Ren et al., 2021) to train the VAE model. Besides, inspired by Prajwal et al. (2020), we independently train a *sync-expert* $D_{sync}$ that measures the possibility that the input audio and 3D landmarks are in-sync, whose training process can be found in Appendix A.2 . The trained sync-expert is then utilized to guide the training of VAE. To summarize, the training loss of our variational motion generator (VG) is as follows:

$$\mathcal{L}_{VG}(\phi, \theta, \epsilon) = -\mathbb{E}_{q_\phi(z|l,a)}[\log p_\theta(l|z, a)] + KL(q_\phi(z|l, a)|p_\epsilon(z|a)) - \mathbb{E}_{\hat{l} \sim p_\theta(l|z,c)}[\log D_{sync}(\hat{l})] \tag{3}$$

where $\phi, \theta, \epsilon$ denote the model parameters of the encoder, decoder and the prior, respectively. $c$ denotes the condition features of VAE. The ground truth and predicted 3D landmarks are represented by $l$ and $\hat{l}$, respectively.

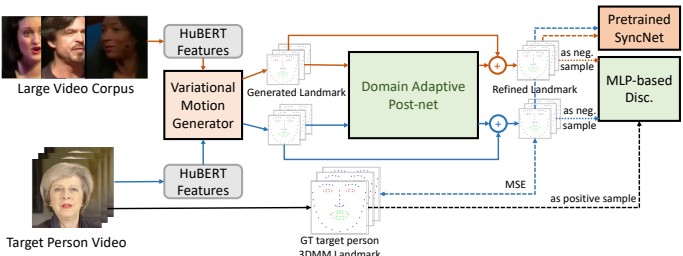

Figure 3: The training process of Domain Adaptative Post-net.

## 3.2 DOMAIN ADAPTIVE POST-NET

As we train the variational motion generator on a large multi-speaker dataset, the model can generalize well with various audio inputs. However, as the scale of the target person video is relatively tiny (about 4-5 minutes) compared with the multi-speaker lip reading dataset (about hundreds of hours), there exists a domain shift between the predicted 3D landmarks and the target person domain. As a consequence, the NeRF-based renderer cannot generalize well with the predicted landmark, which results in blurry or unrealistic rendered images. To this end, A naive solution is to fine-tune the variational generator in the target person dataset. The challenge is that we generally only have a short personalized video, and the generalizability of the model may be lost after the fine-tuning.

Under such circumstances, we design a semi-supervised adversarial training pipeline to perform a domain adaptation. To be specific, we learn a post-net to refine the VAE-predicted 3D landmarks into the personalized domain. We consider two requirements for this mapping: 1) it should preserve the temporal consistency and lip-synchronization of the input sequence; 2) it should correctly map each frame into the target person's domain. To fulfill the first point, we utilize 1D CNN as the structure of post-net and adopt the sync-expert to supervise the lip-synchronization; for the second point, we jointly train an MLP-structured frame-level discriminator that measures the identity similarity of each landmark frame to the target person. The detailed structure of the post-net and discriminator can be found in Appendix A.3.

**Training Process** The training process of post-net is shown in Fig.3. During training, the MLP discriminator tries to distinguish between the ground truth landmark $l'$ extracted from the target person's video and the refined samples $G_\omega(\hat{l})$ generated from the large-scale dataset. We use the LSGAN loss to update the discriminator :

$$\mathcal{L}_{\mathrm{D}}(\delta) = \mathbb{E}_{\hat{l} \sim p_\theta(l|z,c)}[(D_\delta(PN_\omega(\hat{l})) - 0)^2] + \mathbb{E}_{l' \sim p'(l)}[(D_\delta(l') - 1)^2] \qquad (4)$$

where $\omega$ and $\delta$ are the parameters of the post-net $PN$ and discriminator $D$. $l'$ is the ground truth 3DMM landmark of the target person dataset, and $\hat{l}$ is the 3D landmarks refined by the post-net.

As for the training of post-net, the post-net competes with the discriminator while being guided by the pre-trained sync-expert to maintain lip synchronization. Besides, we utilize the target person dataset to provide a weak supervised signal to help the adversarial training. Specifically, we extract the audio $c'$ of the target person video for VAE to predict the landmarks $\hat{l}' \sim p_\theta(l|z,c')$ and encourage the refined landmarks $PN_\omega(\hat{l}')$ to approximate the ground truth expression $l'$. Finally, the training loss of post-net is:

$$\mathcal{L}_{\mathrm{PN}}(\omega) = \mathbb{E}_{\hat{e} \sim p_\theta(l|z,c)}[(D_\delta(PN_\omega(\hat{l})) - 1)^2] + \mathbb{E}_{\hat{l} \sim p_\theta(l|z,c)}[D_{sync}(\hat{l})]$$
$$+ \mathbb{E}_{\hat{l}' \sim p\theta(l|z,c')}[((PN_\omega(\hat{l}') - l')^2] \qquad (5)$$

## 3.3 NERF-BASED RENDERER

We obtain a robust and diverse audio-to-motion mapping through the variational motion generator and post-net. Next, we design a NeRF-based renderer to render high-fidelity frames conditioned on the predicted 3D landmarks.

Figure 4: The training process of NeRF-based renderer.

**3D landmark-conditioned NeRF**    Inspired by Guo et al. (2021), we present a conditional NeRF to represent the dynamic talking head. Apart from viewing direction $d$ and 3D location $x$, the 3D landmarks $l$ will act as the condition to manipulate the color and geometry of the implicitly represented head. Specifically, the implicit function $F$ can be formulated as follows:

$$F_\theta : (x, d, l) \to (c, \sigma) \tag{6}$$

where $c$ and $\sigma$ denote the color and density in the radiance field. To improve the continuity between adjacent frames, we use the 3D landmarks from the three neighboring frames to represent the facial shape, i.e., $l \in \mathbb{R}^{3 \times 204}$. We notice that some facial landmarks only change in a small range, which numerically raises challenges for NeRF to learn the high-frequency image details. Therefore, we normalize the input 3D landmarks **point-wisely**, which is necessary to achieve better visual quality.

Following the setting of volume rendering, to render each pixel, we emit a camera ray $r(t) = o + t \cdot d$ in the radiance field, with camera center $o$, viewing direction $d$. The final color $C$ is calculated by aggregating the color $c$ along the ray:

$$C(r, l; \theta) = \int_{t_n}^{t_f} \sigma_\theta(r(t), l) \cdot c_\theta(r(t), l, d) \cdot T(t) dt \tag{7}$$

where $t_n$ and $t_r$ is the near bound and far bound of ray $r$; $c_\theta$ and $\sigma_\theta$ are the output of the implicit function $F_\theta$, $T(t)$ is the accumulated transmittance along the ray from $t_n$ to $t$, which is defined as:

$$T(t) = \exp(-\int_{t_n}^{t} \sigma_\theta(r(\tau)) d\tau) \tag{8}$$

**Head-aware Torso-NeRF**    To better model the head and torso movement, we train two NeRFs to render the head and torso parts, respectively. As shown in Fig. 4, we first train a head-NeRF to render the head part, then train a torso-NeRF to render the torso part with the rendering image of the head-NeRF as background. Following Guo et al. (2021), we assume the torso part is in canonical space and provide the head pose $h$ to torso-NeRF as a signal to infer the torso movement. The torso-NeRF implicitly learns to expect the location of the rendered head, then rigid the torso from canonical space to render a natural result.

However, this cooperation between head-NeRF and torso-NeRF is fragile since the torso-NeRF cannot observe the head-NeRF's actual output. Consequently, several recent works report that the torso-NeRF produces head-torso separation artifacts (Liu et al., 2022; Yao et al., 2022) when the head pose is relatively large. Based on the analysis above, we propose to provide the torso-NeRF with a perception of the rendering result of the head-NeRF. Specifically, we use the output color $C_{head}$ of the head-NeRF as a **pixel-wise** condition of the torso-NeRF. The torso's implicit function $F_{torso}$ is expressed as:

$$F_{torso} : (x, C_{head}; d_0, \Pi, l) \to (c, \sigma) \tag{9}$$

where $d_0$ is view direction in the canonical space, $\Pi \in \mathbb{R}^{3 \times 4}$ is the head pose that composed of a rotation matrix and a transform vector.

**Training Process**    We extract 3D landmarks from the video frames and use these landmark-image pairs to train our NeRF-based renderer. The optimization target of head-NeRF and torso-NeRF is to reduce the photo-metric reconstruction error between rendered and ground-truth images. Specifically, the loss function can be formulated as:

$$\mathcal{L}_{NeRF}(\theta) = \sum_{r \in \mathcal{R}} ||C_\theta(r, l) - C_g||_2^2 \tag{10}$$

where $\mathcal{R}$ is the set of camera rays, $C_g$ is the color of the ground image.

## 4 EXPERIMENTS

### 4.1 DATASET PREPARATION AND PREPROCESSING

**Dataset preparation.** Our method aims to synthesize high-fidelity talking face images with great generalizability to out-domain audio. To learn robust audio-to-motion mapping, a large-scale lip-reading corpus is needed. Hence we use LRS3-TED (Afouras et al., 2018) to train our variational generator and post-net [2]. Additionally, a certain person's speaking video of a few minutes in length with an audio track is needed to learn a NeRF-based person portrait renderer. To be specific, in order to compare with the state-of-the-art methods, we utilize the data set of Lu et al. (2021) and Guo et al. (2021), which consist of 5 videos of an average length of 6,000 frames in 25 fps.

**Data preprocessing.** As for the audio track, we downsample the speech wave into the sampling rate of 16000 and process it with a pretrained HuBERT model. For the video frames of LRS3 and the target person videos, we resample them into 25 fps and use the toolkit proposed by Deng et al. (2019) to extract the head pose and 3D landmarks. As for the target person videos, they are cropped into 512x512 and each frame is processed with the help of an automatic parsing method (Lee et al., 2020) for segmenting the head and torso part and extracting a clean background.

### 4.2 EXPERIMENTAL SETTINGS

**Comparison baselines.** We compare our GeneFace with several remarkable works: 1) Wav2Lip (Prajwal et al., 2020), which pretrain a sync-expert to improve the lip-synchronization performance; 2) MakeItTalk (Zhou et al., 2020), which also utilize 3D landmark as the action representation; 3) PC-AVS (Zhou et al., 2021), which first modularize the audio-visual representation. 4) LiveSpeech-Portriat (Lu et al., 2021), which achieves photorealistic results at over 30fps; 5) AD-NeRF (Guo et al., 2021), which first utilize NeRF to achieve talking head generation. For Wav2Lip, PC-AVS, and MakeItTalk, the LRS3-TED dataset is used to train the model, and a reference clip of the target person video is used during the inference stage; for LSP, both of LRS3-TED dataset and the target person video is used to train the model; for the NeRF-based method, AD-NeRF, only the target person video is used to train an end-to-end audio-to-image renderer.

**Implementation Details.** We train the GeneFace on 1 NVIDIA RTX 3090 GPU, and the detailed training hyper-parameters of the variational generator, post-net, and NeRF-based are listed in Appendix B. For variational generator and post-net, it takes about 40k and 12k steps to converge (about 12 hours). For the NeRF-based renderer, we train each model for 800k iterations (400k for head and 400k for the torso, respectively), which takes about 72 hours.

### 4.3 QUANTITATIVE EVALUATION

**Evaluation Metrics** We employ the FID score (Heusel et al., 2017) to measure image quality. We utilize the landmark distance (LMD) (Chen et al., 2018) and syncnet confidence score (Prajwal et al., 2020) to evaluate lip synchronization. Furthermore, to evaluate the generalizability, we additionally test all methods with a set of out-of-domain (OOD) audio, which consists of cross-lingual, cross-gender, and singing voice audios.

**Evaluation Results** The results are shown in Table 1. We have the following observations. (1) Our GeneFace achieves good lip-synchronization with high generalizability. Since Wav2Lip is jointly trained with SyncNet, it achieves the highest sync score that is higher than the ground truth video. Our method performs best in LMD and achieves a better sync score than other baselines. When tested with out-of-domain audios, while the sync-score of person-specific methods (LSP and AD-NeRF) significantly drops, GeneFace maintains good performance. (2) Our GeneFace achieves the best visual quality. We observe that one-shot methods (Wav2Lip, MakeItTalk, and PC-AVS) perform

---

[2]we select samples of good quality in the LRS3-TED dataset, the selected subset contains 19,775 short videos from 3,231 speakers and is about 120 hours-long.

| Method | FID ↓ | LMD↓ | Sync ↑ | FID(OOD) ↓ | Sync(OOD) ↑ |
|---|---|---|---|---|---|
| Wav2Lip | 71.40 | 3.988 | **9.212** | 68.05 | **9.645** |
| MakeitTalk | 57.96 | 4.848 | 4.981 | 53.33 | 4.933 |
| PC-AVS | 96.81 | 5.812 | 6.239 | 98.31 | 6.156 |
| LSP | 29.30 | 4.589 | 6.119 | 35.21 | 4.320 |
| AD-NeRF | 27.52 | 4.199 | 4.894 | 35.69 | 4.225 |
| Ground Truth | 0.00 | 0.000 | 8.733 | N/A | N/A |
| GeneFace (ours) | **22.88** | **3.933** | 6.987 | **27.38** | 6.212 |

Table 1: Quantitative evaluation with different methods. Best results are in **bold**.

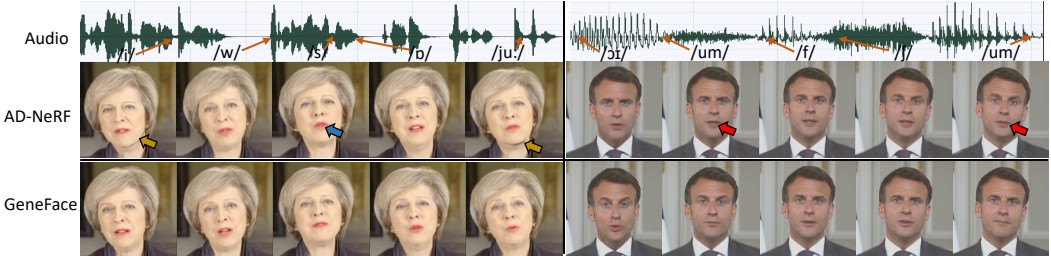

Figure 5: **The comparison of generated key frame results.** We show the phonetic symbol of the key frame and the corresponding synthesized talking heads of AD-NeRF and GeneFace. We mark the head-torso separation artifact, blurry mouth, un-sync results with brown, blue, and red arrow, respectively. Please **zoom in for better visualization**. More qualitative comparisons can be found in demo video.

poorly on FID due to low image fidelity. Since we use 3D landmarks as the condition of the NeRF renderer, it address the mean face problem and leads to better lip syncronization and visual quality than AD-NeRF.

## 4.4 QUALITATIVE EVALUATION

To compare the generated results of each method, we show the keyframes of two clips in Fig.5. Due to space limitations, we only compare our GeneFace with AD-NeRF here and provide full results with all baselines in Appendix C.1. We observe that although both methods manage to generate high-fidelity results, GeneFace solves several problems that AD-NeRF has: 1) head-torso separation (brown arrow) due to the separate generation pipeline of head and torso part; 2) blurry mouth images due to the one-to-many audio-to-lip mapping; 3) unsynchronized lip due to the weak generalizability.

**User Study** We conduct user studies to test the quality of audio-driven portraits. Specifically, we sample 10 audio clips from English, Chinese, and German for all methods to generate the videos, and then involve 20 attendees for user studies. We adopt the Mean Opinion score (MOS) rating protocol for evaluation, which is scaled from 1 to 5. The attendees are required to rate the videos based on three aspects: (1) lip-sync accuracy; (2) video realness; (3) image quality.

We compute the average score for each method, and the results are shown in Table 2. We have the following observations: 1) Our GeneFace achieves comparatively high lip-sync accuracy with Wav2Lip (Prajwal et al., 2020) since both of them learn a generalized audio-to-motion mapping on a large dataset with guidance from a sync-expert. 2) As for the video realness and image quality, the Person-specific methods (LSP, AD-NeRF, and GeneFace) outperform one-shot methods (Wav2Lip, MakeItTalk, and PC-AVS). Although LSP has slightly better image quality than GeneFace, our method achieves the highest video realness and lip-sync accuracy score.

## 4.5 ABLATION STUDY

In this section, we perform ablation study to prove the necessity of each component in GeneFace.

| Methods | Wav2Lip | MakeItTalk | PC-AVS | LSP | AD-NeRF | GeneFace (ours) |
|---|---|---|---|---|---|---|
| Lip-sync Accuracy | 3.77±0.25 | 2.86±0.33 | 3.11±0.30 | 3.65±0.20 | 3.05±0.26 | **3.82±0.24** |
| Image Quality | 3.38±0.19 | 2.84±0.20 | 2.73±0.25 | **3.92±0.13** | 3.44±0.22 | 3.87±0.16 |
| Video Realness | 3.27±0.26 | 2.52±0.30 | 2.46±0.28 | 3.62±0.24 | 3.31±0.24 | **3.87±0.16** |

Table 2: User study with different methods. The error bars are 95% confidence interval.

| Setting | FID↓ | LMD↓ | Sync↑ | FID(OOD)↓ | Sync(OOD)↑ |
|---|---|---|---|---|---|
| GeneFace | **22.88** | **3.933** | **6.987** | **27.38** | **6.212** |
| w/o prior flow | 24.71 | 4.063 | 6.404 | 29.55 | 5.831 |
| w/o sync-expert | 24.02 | 4.151 | 5.972 | 30.77 | 5.549 |
| w/o post-net | 30.26 | 4.532 | 5.085 | 35.58 | 5.248 |
| w. fine-tune | 25.75 | 4.227 | 6.875 | 29.30 | 5.966 |
| w/o head-aware | 26.34 | 3.948 | 6.899 | 28.89 | 6.167 |

Table 3: Ablation study results. The ablation settings are described in Sec. 4.5.

**Varaiational motion generator** We test two settings on the variational motion generator: (1) w/o prior flow, where we replace the flow-based prior with a gaussian prior. The results are shown in Table 3 (line 2), where the Sync score drops by a relatively large margin. This observation suggests that the temporal enhanced latent variable contributes to the stability of the predicted landmark sequence. (2) w/o sync-expert (line 3), where the variational motion generator is no longer supervised by a pretrained sync-expert. We observe that it leads to a significant degradation in Sync score.

**Domain adaptive post-net** In the setting w/o post-net, we remove the domain adaptive post-net, the results are shown in Table 3 (line 4). It can be seen that directly using the 3D landmarks predicted by the variational motion generator leads to a significant performance drop in FID and Sync scores. To further investigate the efficacy of post-net, we utilize T-SNE to visualize the landmarks of different domains in Fig. 10. The visualization results prove that there exists a significant domain gap between the LRS3 dataset and the target person video, and our post-net successfully rigs the predicted landmarks from the LRS3 domain into the target person domain. We also try to replace the post-net with directly fine-tuning on the target person video (line 5), although it achieves a competitive sync score on in-domain audios, its performance in OOD audio is worse.

**Head-aware torso-NeRF** In the w/o head-ware setting, we remove head image condition of the torso-NeRF. The results are shown in Table 3 (line 6). Due to the unawareness of the head's location, the head-torso separation occurs occasionally, which results in a drop in the FID score.

## 5 CONCLUSION

In this paper, we propose GeneFace for talking face generation, which aims to solve the weak generalizability and mean face problem faced by previous NeRF-based methods. A variational motion generator is proposed to construct a generic audio-to-motion mapping based on a large corpus. We then introduce a domain adaptive post-net with an adversarial training pipeline to rig the predicted motion representation into the target person domain. Moreover, a head-aware torso-NeRF is present to address the head-torso separation issue. Extensive experiments show that our method achieves more generalized and high-fidelity talking face generation compared to previous methods. Due to space limitations, we discuss the limitations and future work in Appendix D.

## ACKNOWLEDGMENT

This work was supported in part by the National Natural Science Foundation of China Grant No. 62222211, Zhejiang Electric Power Co.,Ltd.Science and Technology Project No.5211YF22006 and Yiwise.

ETHICS STATEMENT

GeneFace improves the lip synchronization and expressiveness of the synthesized talking head video. With the development of talking face generation techniques, it is much easier for people to synthesize fake videos of arbitrary persons. In most situations, they utilize these techniques to facilitate the movie and entertainment industry and reduce the bandwidth of video streaming by sending audio signals only. However, the talking face generation techniques can be misused. As it is more difficult for people to distinguish synthesized videos, the algorithm may be utilized to spread fake information or obtain illegal profits. Potential solutions like digital face forensics methods to detect deepfakes must be considered. We also plan to include restrictions in the open-source license of the GeneFace project to prevent "deepfake"-related abuse. We hope the public is aware of the potential risks of misusing new techniques.

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

## A  DETAILS OF MODELS

### A.1  VARIATIONAL MOTION GENERATOR

Following PortaSpeech, our variational motion generator consists of an encoder, a decoder, and a flow-based prior model. The encoder, as shown in Fig. 6a, is composed of a 1D-convolution followed by ReLU activation and layer normalization, and a non-causal WaveNet. The decoder, as shown in Fig. 6b, consists of a non-causal WaveNet and a 1D transposed convolution followed by ReLU and layer normalization. The prior model, as shown in Fig. 6c, is a normalizing flow, which is composed of a 1D-convolution coupling layer and a channel-wise flip operation. HuBERT features are utilized as the audio condition of these three modules.

### A.2  SYNC-EXPERT

Our sync-expert inputs a window of $T_l$ consecutive 3D landmark frames and an audio feature clip of size $T_a \times D$, where $T_l$ and $T_a$ are the lengths of the video and audio clip respectively, and $D$ is the dimension of HuBERT features. The sync-expert is trained to discriminate whether the input audio and landmarks are synchronized. It consists of a landmark encoder and an audio encoder, as shown in Fig. 7, both of which are comprised of a stack of 1D-convolutions followed by batch normalization and ReLU. We use cosine-similarity with binary cross-entropy loss to train the sync-expert. Specifically, we compute cosine-similarity for the landmark embedding $l$ and audio embedding $a$ to represent the probability that the input audio-landmark pair is synchronized. The training loss of sync-expert can be represented as:

$$\mathcal{L}_{sync} = CE(\frac{a \cdot l}{\max(||a||_2 \cdot ||l||_2, \epsilon)}) \tag{11}$$

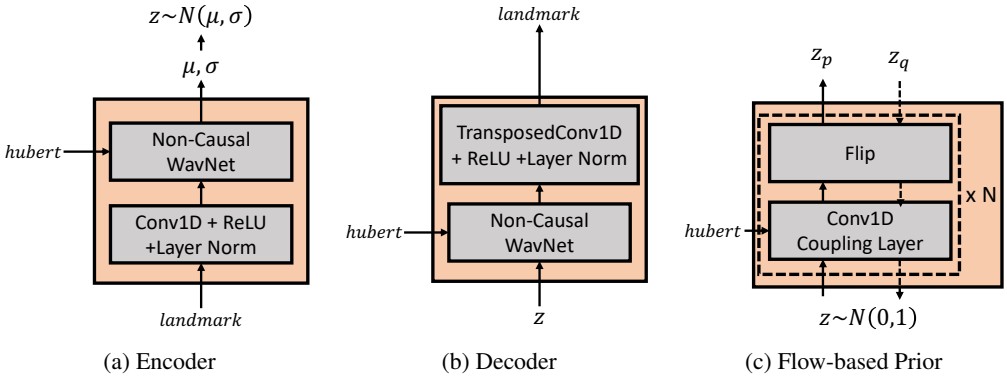

Figure 6: The structure of encoder, decoder, and flow-based prior in variational motion generator.

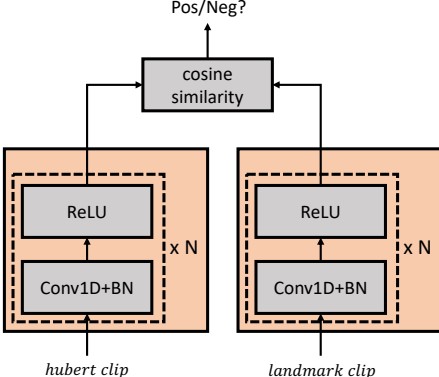

Figure 7: The structure of sync-expert.

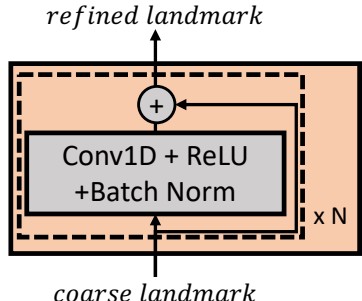

Figure 8: The structure of post-net.

## A.3 DOMAIN ADAPTATIVE POST-NET AND DISCRIMINATOR

The domain adaptative post-net, as shown in Fig. 8, is composed of a stack of residual 1D-convolution followed by ReLU and batch normalization. The discriminator is a MLP composed of a stack of fully connected layers followed with ReLU and dropout.

## B DETAILED EXPERIMENTAL SETTINGS

### B.1 MODEL CONFIGURATIONS

We list the hyper-parameters of GeneFace in Tab. 4.

Table 4: Hyper-parameter list

| Hyper-parameter | | GeneFace |
|---|---|---|
| | Encoder Layers | 8 |
| | Decoder Layers | 4 |
| | Encoder/Decoder Conv1D Kernel | 5 |
| | Encoder/Decoder Conv1D Channel Size | 192 |
| Variational Motion Generator | Latent Size | 16 |
| | Prior Flow Layers | 4 |
| | Prior Flow Conv1D Kernel | 3 |
| | Prior Flow Conv1D Channel Size | 64 |
| | Sync-expert Layers | 14 |
| | Sync-expert Channel Size | 512 |
| | Post-net Layers | 8 |
| | Post-net Conv1D Kernel | 3 |
| Post-net and Discriminator | Post-net Conv1D Channel Size | 256 |
| | Discrimnator Layers | 5 |
| | Discrimnator Linear Hidden Size | 256 |
| | Discrimnator Dropout Rate | 0.25 |
| | Head/Torso-NeRF Layers | 11 |
| NeRF-based Renderer | Head/Torso-NeRF Hidden Size | 256 |
| | Landmark/Head Color Encoder Layers | 3 |
| | Landmark/Head Color Encoder Hidden Size | 128 |

| Setting | L2 error on 3D landmark↓ | LMD↓ |
|---|---|---|
| GeneFace (VAE + Flow + landmark NeRF) | **0.0371** | **3.933** |
| vanilla VAE + landmark NeRF | 0.0385 | 4.063 |
| Regression Model + landmark NeRF | 0.0424 | 4.305 |
| AD-NeRF | N/A | 4.199 |

Table 5: Ablation study on 3D Landmark L2 error.

## C  ADDITIONAL EXPERIMENTS

### C.1  QUALITATIVE RESULTS WITH ALL BASELINES

To compare the generated results of each method, we show the keyframes of one in-domain audio clip in Fig.9. We have the following observations: 1) Wav2Lip achieves competitive lip-sync performance yet generates blurry mouth results; 2) MakeItTalk and PC-AVS fail to preserve the speaker's identity, leading to unrealistic generated results; 3) LSP generates unnatural lip movement during the transition phase of different syllables. Please see our supplementary video for better visualization.

### C.2  EVALUATION ON 3D LANDMARK L2 ERROR

To evaluate the contribution of the variational generator to the quality of the predicted landmark, we adopt L2 error on the predicted 3D landmarks as the metric. We compare our vairiaitonal generator (VAE+Flow) against vanilla VAE and a simple regression model trained with MSE loss. The results are listed in Table 5. It can be seen that removing the prior flow or using a regression-based model leads to a performance drop.

### C.3  T-SNE VISUALIZATION FOR DOMAIN ADAPTATION

To further investigate the efficacy of post-net, we utilize T-SNE to visualize the landmarks of different domains in Fig. 10. The visualization results prove that there exists a significant domain gap between the LRS3 dataset and the target person video, and our post-net successfully rigs the predicted landmarks from the LRS3 domain into the target person domain.

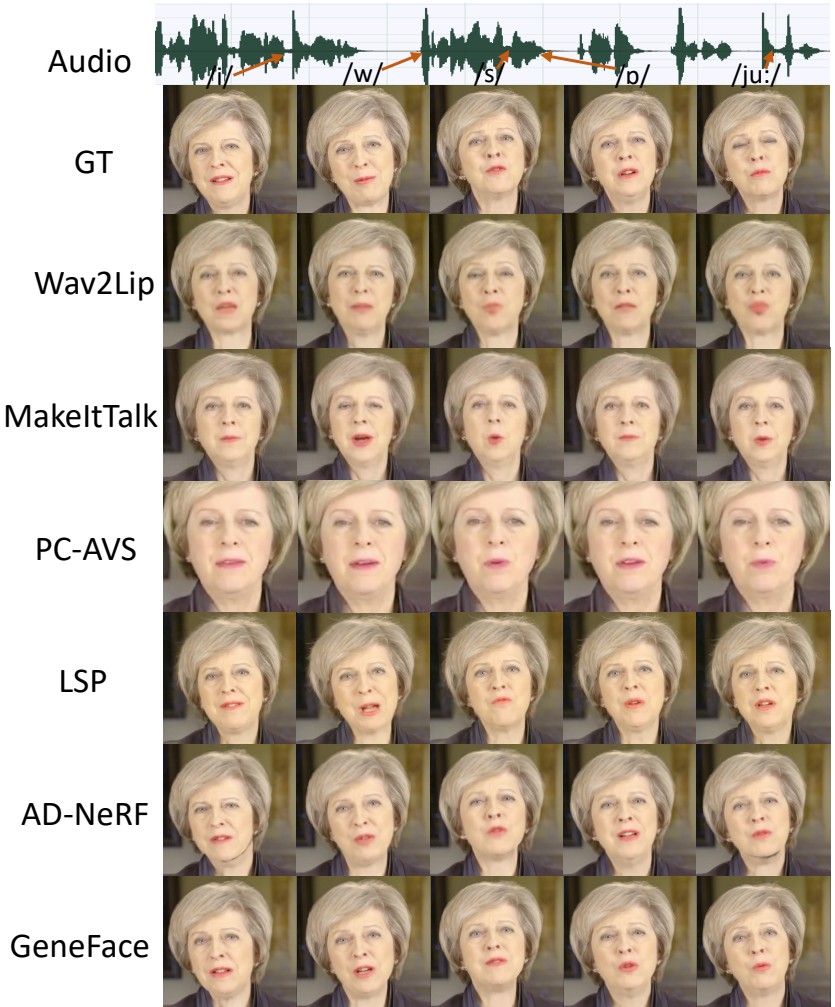

Figure 9: **The comparison of generated key frame results.** We show the phonetic symbol of the key frame and the corresponding synthesized talking heads of all baselines. Please **zoom in for better visualization**. More qualitative comparisons can be found in demo video.

## D   LIMITATIONS AND FUTURE WORK

There are mainly two limitations of the proposed approach. Firstly, we found the landmark sequence generated by variational motion generator and post-net occasionally has tiny fluctuations, which results in some artifacts such as shaking hairs, etc. Currently, we utilize a heuristic post-processing method (Gaussian filter) to alleviate this problem. We will explore better modeling the temporal information in the network architecture to further improve the stability. Secondly, the current NeRF-based renderer is majorly based on the setting of vanilla NeRF, which results in a long training and inference time.

In the future, we will explore three directions: (1) Most importantly, we will try to enhance the performance of the NeRF backend by combining recent progress in accelerated and light-weight NeRF. (2) We will try to utilize GeneFace to achieve style and emotion controllable talking face generation. (3) We will improve the image fidelity and lip-sync quality of GeneFace when driven by singing voices.

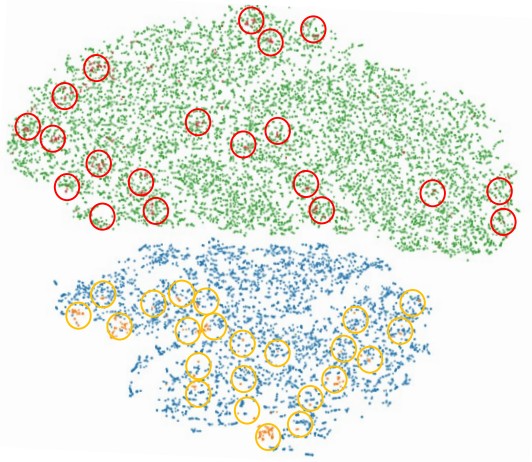

Figure 10: The T-SNE visualization of 3DMM landmarks in different datasets. The green and blue points denote the ground truth landmarks in LRS3 dataset and the target person video; The red and yellow points represent the predicted landmarks without/with the domain adaptation.

