# OpenReview forum: "GeneFace: Generalized and High-Fidelity Audio-Driven 3D Talking Face Synthesis"
_ICLR.cc/2023/Conference — ICLR 2023 poster_

### Official Review · Reviewer_LMgs · 2022-10-23

**Confidence:** 5
**Correctness:** 3
**Technical Novelty And Significance:** 2
**Empirical Novelty And Significance:** 3
**Recommendation:** 5

**Clarity, Quality, Novelty And Reproducibility:**

Clarity: The clarity of this paper is good. The pipeline of the proposed methods is clear and the illustrations are pretty helpful for the readers although there are still lots of typos in the paper.

Quality: The completion of this paper is still good. However the experiments cannot support the motivations and methods very well. The analysis of the proposed methods need to be improved.

Novelty: The novelty is limited. The landmark generation and nerf-based rendering networks are widely adopted for 3D talking head generation. The domain adaptive post-net is interesting to enhance the generalization for the different faces. But lots of works adopt the differences of the target faces and average-face which also helps to enhance the generalization of the methods.

Reproducibility: The clarify of the proposed methods are clear. But the authors did not provide the key code of the method. It is possible to reproduce the performances of the proposed methods but it may be very challenging due to the three complex networks.


**Strength And Weaknesses:**

Strength: 1) The proposed methods achieve remarkable and comparable performances with the previous works;

2) The domain adaptive post-net is interesting for the talking head generation tasks among the different targets. It may help enhance the generalization ability of the generated landmarks for the various faces.

Weaknesses: 1) The novelty is limited. The landmark generation and nerf-based rendering networks are widely adopted for 3D talking head generation. The domain generalization is not firstly introduced for the various faces and audios. The speciality and priority of the proposed methods  need to be highlighted in the paper.

2) The experiments cannot support the motivations and methods very well. The proposed methods aim to enhance the generalization abality for various faces. Comparing with the previous methods on the OOD conditions, the advantages of the proposed methods are not obvious.  Especially the comparisons of these methods on OOD faces need to be analyzed to highlight the effectiveness of the proposed domain adaptive post net.


**Summary Of The Paper:**

This paper proposes GeneFace, a generalized and high-fidelity NeRF-based talking face generation method to generate natural results corresponding to various out-of-domain audio. This paper introduces a variaitional motion generator on a large lip-reading corpus, a domain adaptative post-net to calibrate the result and NeRFbased renderer conditioned on the predicted motion. The proposed methods achieve good performances on the public datasets.

**Summary Of The Review:**

This paper introduces a generalized and high-fidelity NeRF-based talking face generation method for 3D talking head generation. The proposed methods consist of  a variaitional motion generator on a large lip-reading corpus, a domain adaptative post-net to calibrate the result and NeRFbased renderer. The writing and completion of this paper is good. However, this paper is not good enough due to the novelty and experimental weaknesses. The novelty of the proposed methods are limited and the experiments cannot support the motivations of the proposed methods very well.

---

> ### Author Response · Authors · 2022-11-11
> **Author Response to Reviewer LMgs**
>
> We thank the reviewer for the constructive feedback and for considering our work as "of good completion and clarity". We understand that your concerns are mainly related to the paper’s novelty and empirical evaluation. We hope our response resolves your concerns fully.
>
> >  Q1: The novelty is limited. The landmark generation and nerf-based rendering networks are widely adopted for 3D talking head generation. The domain generalization is not firstly introduced for the various faces and audios. **The speciality and priority of the proposed methods need to be highlighted in the paper.**
>
> A1: We would like to summarize our novelty and priority in three points:
>
> 1. We noticed the weak generalizability of previous end-to-end NeRF-based methods to out-of-domain audio, which is an oversight of the community. Motivated by this observation, we propose a three-stage framework that utilizes the 3D landmark as a helpful intermediate, so that the audio-to-motion mapping can be trained on a large dataset and enjoy the generalizability of OOD audio.
> 2. We are also the first work that points out and analyzes the "mean face" problem induced by the one-to-many audio-to-motion mapping in the talking face generation task. To handle the "mean face" problem, we design a generative model (VAE+flow) to alleviate the over-smoothness problem and improve the expressiveness of the generated facial motion.
> 3. We introduce an adversarial domain adaptation pipeline to learn a person-specific post-net. It is the first attempt other than fine-tuning for domain transfer in NeRF-based talking face generation. The combining usage of VAE and post-net denotes that our method could enjoy the diverse and non-over-smoothed results from VAE and target-person-like characteristics brought by post-net.
>
> We have also modified the introduction to highlight the priority of the proposed method, please refer to the **[Revised Part 2]** (which is marked in blue) in the revised manuscript for more details. We hope this reply could address your concerns about the paper's novelty and priority over previous NeRF-based methods.
>
> > Q2: The experiments cannot support the motivations and methods very well. **The proposed methods aim to enhance the generalization abality for various faces.** Comparing with the previous methods on the OOD conditions, the advantages of the proposed methods are not obvious.  Especially the comparisons of these methods on OOD faces need to be analyzed to highlight the effectiveness of the proposed domain adaptive post net.
>
> A2: We suspect that there may have been a minor misunderstanding about the OOD faces and the experiment. For better logicality, we would like to reply in two sub-points:
>
> 1. **About the "generalization ability for various faces"**: Actually, the generalization ability for various faces is achieved by learning a person-specific adaptative post-net for each person. For each person-specific post-net, the task is to calibrate the result into the target person domain. During our experiments, we also noticed the importance to prove the generalization ability for various faces, hence we test the results on target persons with significantly different characteristics (e.g., May, Macron, and Obama). The results show that the proposed method performs well on various faces.
> 2. **About the "motivation" and "the advantages of the proposed methods are not obvious"**: The key motivation of the paper is to improve the robustness of OOD audios in NeRF-based methods by utilizing the large-scale lip reading dataset. It can be seen in the experiment table and our [additional video3](https://geneface.github.io/GeneFace/example_show_improvement.mp4) that our method obviously outperforms AD-NeRF in terms of OOD lip synchronization, while achieving comparable (or slightly better) lip-sync performance to other one-shot methods that also utilize the large dataset (e.g., Wav2Lip, MakeItTalk), which is in line with our expectations. On the other hand, with the benefit of the NeRF-based renderer, we could observe that our method achieves better image quality and video realness than other GAN-based renderers (e.g., Wav2Lip, LSP). An [additional video2](https://geneface.github.io/GeneFace/versus_wav2lip_and_lsp.mp4) which compares the rendering quality can be found on our [demo page](https://geneface.github.io/). Therefore, we believe our experiments support the overall goal of the paper: improving the generalizability of NeRF-based methods to OOD audio while keeping high image fidelity.
>
> We admit that the related discussion in the paper may be not clarified enough, and we have modified these parts for better readability.
>
>
>
> Finally, we appreciate the reviewer’s valuable review and have revised the manuscript accordingly, which we found has improved the clarity and readability of the paper.

---

> ### Author Response · Authors · 2022-11-17
> **Hoping that our response could address your concern**
>
> Dear Reviewer LMgs,
>
> Thank you again for your time and effort in reviewing our work! We would appreciate it if you can let us know if our response has addressed your concern. As the end of the rebuttal phase is approaching, we look forward to hearing from you and remain at your disposal for any further clarification that you might require.
>
> Thanks in advance,
>
> Paper 1601 authors

---

### Official Review · Reviewer_6Jxc · 2022-10-24

**Confidence:** 3
**Correctness:** 3
**Technical Novelty And Significance:** 3
**Empirical Novelty And Significance:** 3
**Recommendation:** 6

**Clarity, Quality, Novelty And Reproducibility:**

Clarity: The paper is clear in the description of the method and the description of the performance metrics.
Quality: Overall quality is above average. The article proposes a framework with some innovations. The experiments are focused on the main issue proposed by this paper.
Novelty: The paper has significant innovation. The domain adaptive post-net is relatively novel, and for the first time, a method other than fine-tuning is proposed for domain transfer in talking face generation.
Reproducibility: The paper has high reproducibility. The description of the method is detailed, and there are detailed experimental details.

**Details Of Ethics Concerns:**

This work might be used to synthesis convincing fake speeches of other individuals.

**Strength And Weaknesses:**

Strength：
1. The paper proposed a novel method to generalize the knowledge learned from the large talking face dataset to out-of-domain data, thus improving few-shot training performance without having to change parameters learned from a large corpus.
2. This paper has done a lot of ablation experiments, and the experimental details shown in Table 3 can enable readers to understand the improvement brought by each method and enhance the reproducibility of the method.
3. There is a detailed description regarding the structure chart of each component and specific hyperparameters of the method in this paper.
Weakness:
1. There is a little description of the dataset used to train the baseline models, e.g., whether the target person videos are used during training. This information is critical for a fair comparison.

**Summary Of The Paper:**

This paper focuses on how to solve the generalization problem of realistic talking face video generation tasks with a landmark refinement module. They proposed GeneFace, a model that introduces additional 3DMM landmark refinement for a target person, thus transferring knowledge learned from a large corpus to out-of-domain data. By utilizing the joint supervision of a lip-syncing expert and an identity discriminator, the model adjusts the generalized 3DMM landmark prediction to match the face of the target person. In this way, the 3DMM prediction can match the distribution of the data used to train the person-specific video renderer. In addition, the paper introduces a head-aware torso-NeRF to address the head-torso separation issue due to dramatic head movements. The generalized and specialized parts of the model are separately trained on a large lip-reading dataset and two smaller lip-reading datasets, and it can achieve good experimental results without fine-tuning, proving its effectiveness.

**Summary Of The Review:**

The paper has good innovation, has done some original work, and the proposed method is concise and has good reproducibility. But the training details of the baseline model are vague, which makes me skeptical about the result of the baseline comparison part.

---

> ### Author Response · Authors · 2022-11-11
> **Author Response to Reviewer 6Jxc**
>
>
>
> We are grateful for your positive review and valuable comments, and we hope our response fully resolves your concerns.
>
> > Q1: There is a little description of the dataset used to train the baseline models, e.g., whether the target person videos are used during training. This information is critical for a fair comparison.
>
> For Wav2Lip, PC-AVS, and MakeItTalk, the LRS3-TED dataset is used to train the model, and a reference clip of the target person video is used during the inference stage; for LSP, both of LRS-TED dataset and the target person video is used to train the model; for the NeRF-based method, AD-NeRF, only the target person video is used to train an end-to-end audio-to-image renderer.
>
> We would like to thank you for this helpful advice since we find introducing the dataset used to train the baseline models is not only crucial for proving a fair comparison but also beneficial to help the reader better understand the goal of our experiment: On the one hand, by additionally utilizing the large dataset in NeRF-based methods, we achieve better lip synchronization than the AD-NeRF baseline; on the other hand, while achieving comparable (or better) sync performance to other baselines that also use the large dataset, our method enjoy better image quality and video realness because of the NeRF-based renderer. We have enriched the related content in the revised manuscript, please refer to **[Revised Part 6]** for more details.
>
> Again, we thank the reviewer for careful and accurate understanding of our paper.

---

> ### Author Response · Authors · 2022-11-17
> **Hoping that our response could address your concern**
>
> Dear Reviewer 6Jxc,
>
> Thank you again for your time and effort in reviewing our work! We would appreciate it if you can let us know if our response has addressed your concern. As the end of the rebuttal phase is approaching, we look forward to hearing from you and remain at your disposal for any further clarification that you might require.
>
> Thanks in advance,
>
> Paper 1601 authors

---

### Official Review · Reviewer_Qesv · 2022-10-24

**Confidence:** 4
**Correctness:** 3
**Technical Novelty And Significance:** 3
**Empirical Novelty And Significance:** 3
**Recommendation:** 8

**Clarity, Quality, Novelty And Reproducibility:**

Although some small mistakes occur in this paper. The writing of the paper is well organized. It is easy for readers to follow. I think this paper is reasonable and reproducible.
The Variational Motion Generator, Domain Adaptative Post-net and the torso-NeRF conditioned on pixel-wise colors are novel in NeRF based talking face system.

**Strength And Weaknesses:**

strengths:
The method could solve one-to-many mapping problem faced by generic NeRF-based talking face systems. And it could solve head-torso separation problem in rendering.

weaknesses:
1) The title use ‘generalized’ to describe this system. It seems this word implies that no subject-specific training is needed. However according to the paper, both the Domain Adaptative Post-net and NeRF renderer adopt subject-specific training to model personalized attributes and need days to converge. I recommend changing this title to avoid misleading.
2) This paper adopts NeRF for high fidelity rendering. This paper should add discussion on NeRF based head techniques such as NerFACE, HeadNeRF and EG3D, but not restricted to NeRF for talking face. One paper is left: Learning Dynamic Facial Radiance Fields for Few-Shot Talking Head Synthesis (ECCV 2022)
3) This paper lacks discussion on limitations and future work.
4) In Fig 1. Variational Motion Generator’s ‘Encoder’ should be ‘Decoder’. In Equ (8), ‘-’ is left in the computing of accumulated transmittance.

**Summary Of The Paper:**

This paper presents a method on audio driven 3D portrait. A three-stage pipeline is given. The first stage trains Variational Motion Generator to predict landmarks based on audio signal. The second stage uses a post-net to compensate for the domain shift between the predicted 3D landmarks and the target person domain. The third stage adopts NeRF for high fidelity rendering conditioned on landmarks. When dealing with torso-NeRF, a pixel-wise condition is added to NeRF function to reduce head-torso separation artifacts.

According to the description in these three stages. The Variational Motion Generator is a generic model. Both the Domain Adaptative Post-net and NeRF renderer adopt subject-specific training to model personalized attributes.

The method could solve one-to-many mapping problem faced by generic NeRF-based talking face systems. And it could solve head-torso separation problem in rendering.

**Summary Of The Review:**

This paper proposes a high-fidelity audio driven 3D talking face system. A novel generalized audio to landmarks module is given. And a subject-specific NeRF module is used to for high fidelity modeling which is an incremental modification on AD-NeRF architecture.
The title didn’t completely suit for the content and reference is not enough. Also limitations and future work should be added.

---

> ### Author Response · Authors · 2022-11-11
> **Author Response to Reviewer Qesv**
>
> Dear reviewer, we sincerely appreciate your careful and accurate understanding of our approach. We hope our response fully resolves your concerns.
>
> >  Q1: The title use ‘generalized’ to describe this system. It seems this word implies that no subject-specific training is needed. However according to the paper, both the Domain Adaptative Post-net and NeRF renderer adopt subject-specific training to model personalized attributes and need days to converge. I recommend changing this title to avoid misleading.
>
> A1:
>
> We agree with your advice. We admit that the word 'generalized' in the title might imply that no subject-specific training is needed. The intention that we use the 'generalized' is to imply that the model is generalizable to out-of-domain audios. Therefore, we consider a new title: **"GeneFace: Towards High-Fidelity Audio-Driven 3D Talking Face Synthesis Generalizable to Out-Of-Domain Audio"**. What do you think about the new title? We also look forward to getting your suggestions.
>
> > Q2: This paper adopts NeRF for high fidelity rendering. This paper should add discussion on NeRF based head techniques such as NerFACE, HeadNeRF and EG3D, but not restricted to NeRF for talking face. One paper is left: Learning Dynamic Facial Radiance Fields for Few-Shot Talking Head Synthesis (ECCV 2022).
>
> A2: Thanks for your helpful comments. We have cited and discussed the mentioned works in the revised manuscript. Please refer to the **[Revised Part 3]** and **[Revised Part 4]** in the related work section for details.
>
> > Q3: This paper lacks discussion on limitations and future work.
>
> A3: Following your advice, we have discussed the limitations and future work in the revised manuscript. And we summarize them here：
>
> - The first limitation is the long training and inference time of the NeRF-based renderer, which we found can be handled by recent works on accelerating NeRF.
> - The second limitation is the occasional unstable results (such as changing hair), we found that the instability is raised in the parallel 3D landmark generation process. Currently, we have alleviated the unstable problem by heuristically post-processing the predicted 3D landmark. In future work, we will explore better modeling the temporal information in the network architecture to further improve the stability.
>
> For more detailed description, please refer to the **[Revised Part 10]** in the revised manuscript.
>
> > Q4: In Fig 1. Variational Motion Generator’s ‘Encoder’ should be ‘Decoder’. In Equ (8), ‘-’ is left in the computing of accumulated transmittance.
>
> A4: Thanks for your careful and accurate reading of our paper. We have fixed the typos in the revised version.
>
>
>
> Following your comments, we have modified the manuscript and we found the clarity and completion of the paper have been improved. Again, we thank the reviewer for the insightful review and "Accept" recommendation for our paper.

---

> ### Author Response · Authors · 2022-11-17
> **Hoping that our response could address your concern**
>
> Dear Reviewer Qesv,
>
> Thank you again for your time and effort in reviewing our work! We would appreciate it if you can let us know if our response has addressed your concern. As the end of the rebuttal phase is approaching, we look forward to hearing from you and remain at your disposal for any further clarification that you might require.
>
> Thanks in advance,
>
> Paper 1601 authors

---

### Official Review · Reviewer_rq8u · 2022-10-24

**Confidence:** 4
**Clarity, Quality, Novelty And Reproducibility:** see comments above.
**Correctness:** 4
**Technical Novelty And Significance:** 3
**Empirical Novelty And Significance:** 2
**Recommendation:** 6

**Strength And Weaknesses:**

**Novelty**

The approach is of minor novelty. Audio-driven talking head synthesis based on NeRF has already been proposed in AD-NeRF and the 3D facial landmark generation is a HuBERT conditioned VAE plus flow modeling. While this specific architecture has not been used before, the paper does not provide significant novelty beyond modified architectures and details in the training.

**Technical Details**

The technical presentation of the paper seems sound and is easy to follow. Details on the model are explained in the appendix, such that I believe the method can be reproduced from the information given in the paper.

**Empirical Evaluation**

(a) Qualitative results (video linked in the paper): The output of the approach produces significant artifacts such as hair that is constantly changing it's size/volume, unstable and wobbly face reconstruction, and significant artifacts in the neck area. I do not agree that this is an improvement over MakeItTalk, PC-AVS, LSP, or AD-NeRF.

(b) User study. The user study has a small sample size (5 clips from 3 languages, 10 attendees) and given this small sample size, the differences in MOS scores compared to existing work seems not statistically significant. I'd further advise to include the standard deviation to the mean opinion scores, which will provide a confidence on the ratings.

(c) Quantitative Evaluation. All metrics are neural feature maps. These alone can be delusive, a metric comparison would be more convincing. One option would be to compare the l2 error on facial landmarks to evaluate accuracy of lip and general face animation.

(d) In-depth analysis. One major contribution of this paper is the variational motion generator for the generation of 3D facial landmarks. However, this contribution is lacking empirical analysis. Fig. 6 provides some insights on the domain adaptation, but a proper evaluation of the quality of the produced landmarks is not given. It would be more convincing if the authors could demonstrate clear and consistent improvements in 3D landmark generation by their variational motion model compared to a simple regression baseline, a simple VAE baseline, and landmarks from other approaches. Unfortunately, neither a quantitative evaluation (l2 error on predicted landmarks) nor a qualitative evaluation is provided.

**Summary Of The Paper:**

The authors address the problem of synthesizing realistic talking faces from audio. They build upon a VAE-style encodings of 3D facial landmarks that are created from HuBERT features extracted from the audio signals. Instead of a vanilla VAE, the authors argue that a flow-based (non-Gaussian) prior is beneficial for better landmark generation. Conditioned on the predicted facial landmarks, a NeRF-style renderer then synthesizes an animated head and torso.

**Summary Of The Review:**

The paper is of minor novelty and the main technical contribution lacks in-depth evaluation. The presented qualitative results are not convincing, i.e. they do not seem to be better than existing approaches to me.

**Edit after rebuttal:**

The authors provided a strong and convincing rebuttal. They were able to address the major points that, in my view, put the paper below acceptance threshold. The authors could demonstrate that with appropriate post-processing, their method outperforms AD-NeRF, They also provided further evaluation that closes the gap the paper had before.

I consider this a substantial improvement to the original paper and therefore will upgrade my rating.

---

> ### Author Response · Authors · 2022-11-11
> **Author Response to Reviewer rq8u (Part 1/3)**
>
> We thank the reviewer for the constructive feedback and considering our work as technically sound. We hope our response resolves your concerns fully.
>
> >  Q1: **Novelty** The approach is of minor novelty. Audio-driven talking head synthesis based on NeRF has already been proposed in AD-NeRF and the 3D facial landmark generation is a HuBERT conditioned VAE plus flow modeling. While this specific architecture has not been used before, the paper does not provide significant novelty beyond modified architectures and details in the training.
>
> A1:
>
> We would like to reply to your concerns and highlight the novelty of the approach in three points.
>
> 1. Although AD-NeRF first learns an audio-to-image mapping based on NeRF, we found its weakness in generalizability to OOD audios. Therefore, our work first proposes to utilize the 3D landmark as a helpful intermediate, so that the audio-to-motion mapping can be trained on a large dataset and enjoy the generalizability to OOD audios. As current NeRF-based methods mainly focus on improving image quality, we believe this work could pave the way for more generalized NeRF-based talking face generation.
> 2. After deciding to learn an audio-to-motion model independent from the NeRF-based renderer, we notice the "mean face" problem in previous works, which is induced by the one-to-many audio-to-motion mapping in the talking face generation task. Motivated by this observation, the specific generative model architecture, a HuBERT conditioned VAE plus normalizing flow is introduced to alleviate the over-smoothness and improve the expressiveness of the generated facial motion. The proposed VAE+Flow model, shows better performance in lip-sync than the regression-based model and other baselines, as we will discuss in detail in **A5**.
> 3. We also introduce an adversarial domain adaptation pipeline to learn a person-specific post-net. It is the first attempt other than fine-tuning for domain transfer in NeRF-based talking face generation. The combining usage of VAE and post-net denotes that our method could enjoy the diverse and non-over-smoothed results from VAE and target-person-like characteristics brought by post-net.
>
> To better highlight the contributions and novelty, we have also modified the Introduction in the revised manuscript to highlight the priority of the proposed method, please refer to **[Revised Part 2]** for more details. We hope this reply could address your concerns about the paper's novelty over previous NeRF-based methods.
>
> > Q2: Qualitative results (video linked in the paper): The output of the approach produces significant artifacts such as hair that is constantly changing it's size/volume, unstable and wobbly face reconstruction, and significant artifacts in the neck area. I do not agree that this is an improvement over MakeItTalk, PC-AVS, LSP, or AD-NeRF.
>
> A2:
>
> Thanks for your helpful feedback. As for the produced artifacts, we admit that there is an unstable problem in the demo video, and we suspect it is due to the instability in the predicted 3D landmarks. In our framework, to improve efficiency, the whole landmark sequence is produced in a single forward of the variational generator and post-net, which may induce temporal fluctuation. We have alleviated this problem by post-processing (gaussian filter) the predicted landmark. Please watch the **additional Video 1** at [this link ](https://geneface.github.io/GeneFace/before_after_postprocessing.mp4)on our demo page for the results after using the post-processing method. We noticed that there are significantly fewer artifacts in the current stabilized GeneFace. And the comparison between our stabilized GeneFace and the baselines is shown in the **additional Video 2** at [this link](https://geneface.github.io/GeneFace/versus_wav2lip_and_lsp.mp4), in which we can see that GeneFace has the significantly better image quality and video realness than Wav2Lip and LSP. We leave a more elegant way to improve temporal stability as future works.
>
> To better show our lip-sync performance improvement over the major baseline, AD-NeRF, we carefully compare the lip synchronization results of AD-NeRF and GeneFace in the **additional Video 3** at [this link](https://geneface.github.io/GeneFace/example_show_improvement.mp4). We can see that GeneFace has obviously better lip synchronization performance than AD-NeRF when driven by OOD audios. By contrast, AD-NeRF tends to half-close the mouth and the mouth is relatively more blurry than GeneFace, which is identified as the "mean face" problem due to fitting the one-to-many audio-to-image mapping with a regression-based model.
>
> Due to space limitations, please refer to the next comment for our replys to other questions.

---

> > ### Author Response · Authors · 2022-11-11
> > **Author Response to Reviewer rq8u (Part 2/3)**
> >
> > > Q3: User study. The user study has a small sample size (5 clips from 3 languages, 10 attendees) and given this small sample size, the differences in MOS scores compared to existing work seems not statistically significant. I'd further advise to include the standard deviation to the mean opinion scores, which will provide a confidence on the ratings.
> >
> > A3: Thanks for your advice. We have added 5 tested clips and invited additional 10 attendees to give the rating. So that each case has 200 samples. The mean and 95% confidence interval of MOS results are listed as follows. **Table 2** in the manuscript is modified accordingly.
> >
> > |                    | Wav2Lip       | MakeItTalk    | PC-AVS        | LSP               | AD-NeRF       | GeneFace (ours)   |
> > | ------------------ | ------------- | ------------- | ------------- | ----------------- | ------------- | ----------------- |
> > | Image Quality      | 3.38$\pm$0.19 | 2.84$\pm$0.20 | 2.73$\pm$0.25 | **3.92$\pm$0.13** | 3.44$\pm$0.22 | 3.87$\pm$0.16     |
> > | Lip Syncronization | 3.77$\pm$0.25 | 2.86$\pm$0.33 | 3.11$\pm$0.30 | 3.65$\pm$0.20     | 3.05$\pm$0.26 | **3.82$\pm$0.24** |
> > | Video Realness     | 3.27$\pm$0.26 | 2.52$\pm$0.30 | 2.46$\pm$0.28 | 3.62$\pm$0.24     | 3.31$\pm$0.24 | **3.87$\pm$0.16** |
> >
> > > Q4: Quantitative Evaluation. All metrics are neural feature maps. These alone can be delusive, a metric comparison would be more convincing. One option would be to compare the l2 error on facial landmarks to evaluate accuracy of lip and general face animation.
> >
> > A4:
> >
> > Thank you for pointing out this problem. The reason why we did not choose L2 error or other metrics in the original manuscript is that we have noticed the one-to-many mapping problem in audio-to-motion, which means that there may be several reasonable GT motions for the same audio input. As a result, reconstruction metrics such as L2 error may not capture fine details in the generated sample and give high scores to samples with unsatisfying over-smoothness results. Therefore, in the original manuscript, we only use the neural-feature-based metrics as they are more correlated with human judgment. However, we admit that it is necessary to provide a non-neural-feature-based metric comparison.
> >
> > Following your advice, we choose Landmark Distance (LMD) from [1] to evaluate lip synchronization. Because we use the variational generator for landmark generation, we run our method 5 times and calculate the mean LMD. By contrast, other algorithms are regression-based so only 1 run is needed. The final LMD results are as follows. We can see that GeneFace performs best in LMD. **Table 1** in the manuscript is modified accordingly.
> >
> > |      | Wav2Lip | MakeItTalk | PC-AVS | LSP   | AD-NeRF | GeneFace |
> > | ---- | ------- | ---------- | ------ | ----- | ------- | -------- |
> > | LMD  | 3.988   | 4.848      | 5.812  | 4.589 | 4.199   | 3.933    |
> >
> > Due to space limitations, please refer to the next comment for our replys to other questions.

---

> > > ### Author Response · Authors · 2022-11-11
> > > **Author Response to Reviewer rq8u (Part 3/3)**
> > >
> > >
> > >
> > > > Q5:  In-depth analysis. One major contribution of this paper is the variational motion generator for the generation of 3D facial landmarks. However, this contribution is lacking empirical analysis. Fig. 6 provides some insights on the domain adaptation, but a proper evaluation of the quality of the produced landmarks is not given. It would be more convincing if the authors could demonstrate clear and consistent improvements in 3D landmark generation by their variational motion model compared to a simple regression baseline, a simple VAE baseline, and landmarks from other approaches. Unfortunately, neither a quantitative evaluation (l2 error on predicted landmarks) nor a qualitative evaluation is provided.
> > >
> > > A5: Thanks for your suggestion, to better analyze the contribution of our variational generator, we use L2 error on the predicted 3D landmarks as the metric. Then we compare our variational generator (VAE+Flow) against vanilla VAE and a simple regression model trained with MSE loss. The results are listed in the table below. It can be seen that VAE+Flow performs the best on the L2 error. We also provide [**Additional Video4**](https://geneface.github.io/GeneFace/display_3d_landmark.mp4) on our demo page for better comparison.
> > >
> > > Besides, we also compute the LMD on the final rendered video. It can be seen that VAE+Flow still performs the best. An interesting finding is that (the regression model + landmark NeRF) performs worse than the end-to-end AD-NeRF. We suspect it is because the performance loss is accumulated through these two independent regression-based models.
> > >
> > > We have added this additional ablation study in the revised manuscript, please refer to **[Revised part 9]** for more details.
> > >
> > > | Method                                | L2 error on 3D landmark | LMD       |
> > > | ------------------------------------- | ----------------------- | --------- |
> > > | GeneFace (VAE + Flow + landmark NeRF) | **0.0371**              | **3.933** |
> > > | vanilla VAE + landmark NeRF           | 0.0385                  | 4.063     |
> > > | Regression Model + landmark NeRF      | 0.0424                  | 4.305     |
> > > | AD-NeRF                               | /                       | **4.199** |
> > >
> > >
> > >
> > > # Summary
> > > In summary, following the given comments, we have performed several additional experiments and revised the manuscript a lot, which have improved the soundness and depth of the paper. Again, we would like to appreciate the reviewer’s valuable review. We sincerely hope the reviewer will reconsider their rating in light of the rebuttal and new experiments.
> > >
> > >
> > >
> > > [1] Chen L, Li Z, Maddox R K, et al. Lip movements generation at a glance[C]//Proceedings of the European Conference on Computer Vision (ECCV). 2018: 520-535.

---

> ### Author Response · Authors · 2022-11-14
> **Thanks for your acknowledgment of our rebuttal**
>
> Dear Reviewer,
>
> Thanks for your acknowledgment of our rebuttal. Again, we would like to thank you for your deep understanding of the paper and insightful comments, which have improved the soundness and depth of the paper by a large margin.
>
> Paper 1601 Authors

---

### Official Review · Reviewer_vzEy · 2022-10-25

**Confidence:** 3
**Correctness:** 4
**Technical Novelty And Significance:** 3
**Empirical Novelty And Significance:** 3
**Recommendation:** 6

**Clarity, Quality, Novelty And Reproducibility:**

This paper has well-organized structure, and it is clear in logicality. This work proposed three sub-network and they can solve problems. It is reproducible work according to the designed network structure and experiment results.

**Details Of Ethics Concerns:**

Use celebrity face data

**Strength And Weaknesses:**

Strength:
This paper has well-organized structure, and it is clear in logicality. This work has well-designed network structure and proposed structure, which can actually solve proposed problems. This work combines previous works and gives an new approach to generating talking face in another way, which moves the NeRF-based method field forward a bit.

Weaknesses:
Some evaluation metric do not outperform other methods, this work could design more evaluation metrics to evaluate the results.
The generated face in the demo video is moving around, maybe consider temporal information and design into network structure.
The lip-synchronization is not as good as wav2lip, the synchronization can be improved.

**Summary Of The Paper:**

This paper proposed an end-to-end NeRF-based method, for talking face generation. It was trying to solve 2 problems: 1) weak generalizability due to small scale of training data, 2) “mean face” result: bad image quality and bad lip-synchronization. This paper proposed 3 parts: 1) Variational Motion Generator used to generate landmarks from audio, 2) Domain Adaptative Post-Net for landmark refinement, and 3) NeRF based Renderer for final frame generation. It produces good talking face generation results.

**Summary Of The Review:**

The paper is overall well written and much of it is well described. It combines previous works and proposes an new approach to generating 3D talking face. It produces better results than previous NeRF-based work. I recommend this work.

---

> ### Author Response · Authors · 2022-11-11
> **Author Response to Reviewer vzEy**
>
> We are grateful for your positive review and valuable comments, and we hope our response fully resolves your concerns.
>
> >  Q1: Some evaluation metric do not outperform other methods, this work could design more evaluation metrics to evaluate the results.
>
> A1: Thanks for your advice. As can be seen in Table 1 in the paper, Wav2Lip achieves better SyncScore (9.212) than ground truth data (8.733) and GeneFace (6.987). This may be because the SyncScore is a neural metric obtained by a pretrained SyncNet[1], which is also utilized during the training of Wav2Lip. Therefore, to better measure the lip synchronization, we adopt Landmark Distance (LMD) proposed in [2], which can be represented as :
> $$
> LMD = \frac{1}{T}\times \frac{1}{P}\Sigma_t\Sigma_p ||l_{t,p}-\hat{l}_{t,p}||
> $$
> where $l$ is the normalized 2D landmark extracted from the final rendered images, $t$ and $p$ denote timestep and key point index, respectively. The LMD results are listed as follows. We found that GeneFace slightly outperforms Wav2Lip. **Table 1** in the manuscript is modified accordingly.
>
> |      | Wav2Lip | MakeItTalk | PC-AVS | LSP   | AD-NeRF | GeneFace |
> | ---- | ------- | ---------- | ------ | ----- | ------- | -------- |
> | LMD  | 3.988   | 4.848      | 5.812  | 4.589 | 4.199   | 3.933    |
>
> > Q2: The generated face in the demo video is moving around, maybe consider temporal information and design into network structure.
>
> A2: Thanks for your helpful comments. We have also noticed that there was a temporal instability problem in the previous demo video. We think this is because the current variational generator and post-net generate the whole landmark sequence in one forward, rather than only generating one segment of a fixed window length, which is a common trick in previous works.
>
> To address this temporal instability problem, we tried several methods and found a heuristic signal post-processing method (gaussian filter to the predicted landmark sequence) partially handles the problem. you can also see the **additional Video 1** at [this link ](https://geneface.github.io/GeneFace/before_after_postprocessing.mp4)in our demo page for better comparison. In the future, we will explore considering temporal information and designing into network structure for more stable results. We have listed this problem in future work in the revised manuscript, as can be seen in **[Revised Part 10]** in the revised manuscript.
>
> > Q3: The lip-synchronization is not as good as wav2lip, the synchronization can be improved.
>
> A3: We agree that the synchronization of GeneFace can be further improved. Ideally, our variational generator could achieve comparable lip-sync performance to Wav2Lip, since both of these two models are trained on the LRS3 dataset and learned from a pre-contrastively-learned sync-expert. However, we have to additionally project the predicted landmark into the target person domain to guarantee that the nerf-based renderer could properly render the high-quality image. We think this domain adaptative transform might induce a performance drop in lip synchronization. Still, as we mentioned in **A1**, from the perspective of the LMD metric, our methods slightly outperform Wav2Lip. To summarize, we admit that there is a trade-off between lip-sync and image quality in designing the talking face generation system. We will explore better solutions to improve lip synchronization while achieving high image fidelity.
>
> Again, we thank the reviewer for the insightful review and positive recommendation for our paper.
>
>
>
> [1] Chung J S, Zisserman A. Out of time: automated lip sync in the wild[C]//Asian conference on computer vision. Springer, Cham, 2016: 251-263.
>
> [2] Chen L, Li Z, Maddox R K, et al. Lip movements generation at a glance[C]//Proceedings of the European Conference on Computer Vision (ECCV). 2018: 520-535.

---

> ### Author Response · Authors · 2022-11-17
> **Hoping that our response could address your concern**
>
> Dear Reviewer vzEy,
>
> Thank you again for your time and effort in reviewing our work! We would appreciate it if you can let us know if our response has addressed your concern. As the end of the rebuttal phase is approaching, we look forward to hearing from you and remain at your disposal for any further clarification that you might require.
>
> Thanks in advance,
>
> Paper 1601 authors

---

### Author Response · Authors · 2022-11-11
**General Response to All Reviewers**

We would like to thank the reviewers for their constructive reviews! Following the comments and suggestions of reviews, we have revised the manuscript, and the revised parts are marked in blue. Here we summarize the revision as follows:

- To handle the unstable problem (eg., shaking hairs) pointed out by Reviewer vzEy and Reviewer rq8U, we adopt a post-processing method to the predicted landmark sequence, which significantly reduces the artifacts. Please see the [Additional Video1](https://geneface.github.io/GeneFace/before_after_postprocessing.mp4) on our demo page for better comparison.
- To better show the performance improvement of the proposed method, we provide [Additional Video2](https://geneface.github.io/GeneFace/versus_wav2lip_and_lsp.mp4) to show the improved image quality and video realness, and [Additional Video3](https://geneface.github.io/GeneFace/example_show_improvement.mp4) to show the improved lip synchronization to OOD audio.
- In section 1, we highlight the novelty of the proposed approach, as suggested by Reviewer rq8U and Reviewer LMgs.
- In section 2, we enrich the related work about NeRF-based face rendering, as suggested by Reviewer Qesv.
- In section 4.2, we add description of the dataset used to train the baseline models, as suggested by Reviewer 6Jxc.
- In section 4.3, we additionally test the Landmark Distance (LMD) as a non-neural-map-metric to measure the lip synchronization, as suggested by Reviewer vzEy and Reviewer rq8u.
- In section 4.4, we enlarge the scale of user study, and calculate the 95% confidence interval for MOS, as suggested by Reviewer rq8u.
- In section 4.5, we add a ablation study to evaluate the contribution of the variational generator to generate high-quality 3D landmarks, as suggested by Reviewer rq8u. Please see the [Additional Video4](https://geneface.github.io/GeneFace/display_3d_landmark.mp4) for better comparison.
- In section 5, we discuss the limitations and future work, as suggested by Reviewer Qesv.

Thanks again for the reviewers' great efforts and valuable comments, which have improved the soundness of the manuscript. We have carefully addressed the main concerns and provided detailed responses to each reviewer. We hope you might find the responses satisfactory. We would be grateful if we could hear your feedback regarding our answers to the reviews.

---

### Author Response · Authors · 2022-11-14
**Dear AC and Reviewers,**

Thanks again for your great efforts and valuable comments.

We have carefully addressed the main concerns and provided detailed responses to each reviewer. We hope you might find the responses satisfactory. As the end of the rebuttal phase is approaching, we would be grateful if we could hear your feedback regarding our answers to the reviews. We will be very happy to clarify any remaining points (if any).

Thanks in advance, Paper 1601 authors

---

### Author Response · Authors · 2022-11-18
**Dear AC and reviewers,**

As the end of the rebuttal phase is approaching, to follow the Author Guide, we have changed the color of the revised parts (which are previously marked in blue) in the manuscript to black and removed the labels like **[Revised Part 1]**.

For your convenience, you can still easily access the previous revised version at [this link](https://openreview.net/references/pdf?id=akJY0vQ51n), in which the revised parts are highlighted.

Paper 1601 authors

---

### Decision · Program_Chairs · 2023-01-20

**Decision:**

Accept: poster

**Justification For Why Not Higher Score:**

The main reasons are due to the weakness listed in the summary, especially the domain adaptive transform associated with posenet might reduce the quality of Lip-synchronization while improving image fidelity, and the training and inference of NeRF-based renderer are time consuming.

**Justification For Why Not Lower Score:**

The proposed DNN architecture is reasonable and addresses two issues with previous NeRF-based method -- OOD audio generalization and "mean face" problem.
Experiments demonstrate the effectiveness of the proposed method for generating high-fidelity talking face comparing with previous methods.


**Metareview: Summary, Strengths And Weaknesses:**

Summary:
This paper presents a NeRF-based method GeneFace for generating realistic talking face from audio. The authors introduce a VAE encoder and a domain-adaptive postnet for mapping audio to 3D landmarks, and for improving generalization to out-of-domain audio.
Experiments demonstrate the effectiveness of the proposed method for generating high-fidelity talking face comparing with previous methods.

Strengths:
- The proposed DNN architecture is reasonable and addresses two issues with previous NeRF-based method -- OOD audio generalization and "mean face" problem.
- The main innovation is the domain adaptive postnet, a method other than fine-tuning for domain transfer in talking face generation.
- The DNN model is described in detail, and the results could be reproducible.
- The paper is well-written and easy to follow.


Weaknesses:
- The domain adaptive transform associated with posenet might reduce the quality of Lip-synchronization while improving image fidelity.
- There is a temporal instability problem with GeneFace and might introduce artifacts such as shaking hairs.
- The training and inference of NeRF-based renderer are time consuming.


**Note From Pc:**

if the above contains the word "oral" or "spotlight" please see: "oral" presentation means -> notable-top-5% and "spotlight" means -> notable-top-25%. As stated in our emails, we are disassociating presentation type from AC recommendations